# Invasive Hypervirulent *Klebsiella pneumoniae* Syndrome Originating from an Anorectal Abscess as Opposed to a Pyogenic Liver Abscess

**DOI:** 10.3390/medicina58101450

**Published:** 2022-10-14

**Authors:** Kum Ju Chae, Jaehyeon Lee, Joo-Hee Hwang, Jeong-Hwan Hwang

**Affiliations:** 1Department of Radiology, Jeonbuk National University Medical School and Hospital, Jeonju 54907, Korea; 2Research Institute of Clinical Medicine, Jeonbuk National University-Biomedical Research Institute of Jeonbuk National University Hospital, Jeonju 54907, Korea; 3Department of Laboratory Medicine, Jeonbuk National University Medical School and Hospital, Jeonju 54907, Korea; 4Department of Internal Medicine, Jeonbuk National University Medical School and Hospital, Jeonju 54907, Korea

**Keywords:** hypervirulence, *Klebsiella pneumoniae*, anorectal abscess

## Abstract

An immunocompetent 49-year-old man presented with swelling and pain in the lower region of his left leg that had lasted for 4 weeks. The diagnosis was severe pyomyositis and osteomyelitis in the lower left leg caused by hypervirulent *Klebsiella pneumoniae* (hvKP) along with multiple metastatic infections in the kidneys, lungs, and brain originating from an anorectal abscess. A virulence-gene analysis revealed that the isolated *K. pneumoniae* harbored *rmpA*, *entB*, *ybtS*, *kfu*, *iutA*, *mrkD*, and *allS*-virulence genes and belonged to the K1 capsular serotype. After repeated abscess drainage procedures, intravenous ceftriaxone was administered for more than 10 weeks, and the patient’s infection was controlled. We focused on the clinical features of hvKP originating from an anorectal abscess without a pyogenic liver abscess. We suggest that hvKP be considered a causative pathogen of pyomyositis and osteomyelitis resulting in multiple metastatic infections in an immunocompetent patient, and more information on the unexpected multiple metastatic infections should be obtained from a virulence analysis of *K. pneumoniae*.

## 1. Introduction

Pyomyositis is a purulent infection of the skeletal muscles that arises from the hematogenous spread of bacteria and is usually associated with abscess formation [1]. Up to 90% of cases can be attributed to gram-positive cocci such as *Staphylococcus aureus,* [2] with gram-negative enteric bacilli accounting for only a small portion of most of the remainder (9.6%) of all causative pathogens [3,4]. *Klebsiella* species account for 22.7% of gram-negative bacterial pyomyositis cases and are primarily detected in immunocompromised patients [3,5].

Since Friedlander’s first report of *K. pneumoniae* infection in 1882, the main causative agent has been recognized as classic *K. pneumoniae* (cKP) [5]. However, since the first case of septic endophthalmitis associated with a pyogenic liver abscess caused by *K. pneumoniae* was reported in Taiwan in 1986, hypervirulent *K. pneumoniae* (hvKP) has spread to become a global pathogen [6]. *K. pneumoniae* is divided into two categories based on its clinical and bacterial phenotypic features: cKP and hvKP [7]. hvKP differs from cKP in its clinical and phenotypic characteristics [5]. cKP strains have been associated with infections involving the urinary tract, lungs, abdominal cavity, intravascular regions, surgical sites, and soft tissue and have been identified as the cause of subsequent bacteremia in hospitals and long-term care facilities [7]. Metastatic infections caused by enteric gram-negative bacilli are uncommon without predisposing factors such as neutropenia or malignancy [5]. However, hvKP causes a range of severe metastatic infections in various parts of the body, regardless of the underlying conditions [7]. At least 78 capsular polysaccharide serotypes exist in *K. pneumoniae* [5]. For hvKP strains, eight capsular serotypes have been described to date, namely, K1, K2, K5, K16, K20, K54, K57, and KN1 [5]. According to recent reports, the K1 and K2 serotypes are mainly associated with hvKP, and virulence factors, such as *rmpA*, aerobactin, *kfu*, and *allS*, are more dominant in hvKP than in cKP [7].

Herein, we report a life-threatening case of hvKP with a metastatic infection throughout the body from an anorectal abscess that manifested as pyomyositis and osteomyelitis in the lower extremities of an otherwise healthy adult. Our study aimed to investigate the virulence factors associated with hvKP and to report on hvKP as a causative factor of community-onset pyomyositis and osteomyelitis resulting in metastatic infections in an immunocompetent patient.

## 2. Case Presentation

A 49-year-old man without any history of underlying disease presented to the emergency room with swelling and pain in his lower left leg. Pain and swelling had started 4 weeks earlier. These symptoms gradually worsened, and a fever developed 3 days prior to the visit. A physical examination of the left leg revealed swelling, tenderness, erythema, and a sensation of heat. The following laboratory results were obtained: white blood cell count, 9410 cells/μL (neutrophil 90%); hemoglobin, 6.4 g/dL; platelet count, 124,000 cells/μL; total protein, 5.0 g/dL; albumin, 2.1 g/dL; blood urea nitrogen, 57 mg/dL; creatinine, 0.83 mg/dL; procalcitonin, 13.8 μg/L; and C-reactive protein, 18.6 mg/dL. The HIV antibody test was negative, and the levels of immunoglobulins (Ig) M, IgA, and IgG and complements were within the normal range. Radiography and magnetic resonance imaging (MRI) of the left leg revealed a large abscess running through all the thigh muscles, accompanied by osteomyelitis of the left femur (Figure 1). Chest and abdominal computed tomography (CT) scans were performed owing to the tenderness around the perineum, and they showed anorectal abscesses, microabscesses in both kidneys, and septic embolism in both lungs (Figure 2A–C). However, there was no evidence of liver abscess. Empirical ceftriaxone and metronidazole were administered intravenously (IV). The patient underwent surgical debridement and drainage of the abscesses in both the anorectal area and lower left leg, and >500 mL of pus was removed. The following day, the patient experienced a sudden seizure, and a brain diffusion MRI revealed a multifocal cerebral infarction resulting from a septic embolism (Figure 2D).

The infection was controlled with 10 weeks of intravenous ceftriaxone administrations and repeated abscess drainage. Although the patient was treated with appropriate antibiotics for a long duration, severe bone damage from osteomyelitis and loss of a significant amount of lower leg muscle were observed. In addition, this patient developed a left femur shaft fracture during admission but could not undergo internal fixation for the fracture due to his poor general condition. The patient was transferred to a nursing care hospital in a bedridden state and did not recover functionality in the left leg because of osteomyelitis and an accompanying pathological fracture.

### Microbiological Evaluation

*K. pneumoniae* was identified in a pus specimen obtained from the operating room using VitekMS (bioMérieux Inc., Marcy-l’Étoile, France). Blood cultures taken at the emergency room and during hospitalization were negative. Vitek 2 (bioMérieux Inc., Hazelwood, MO, USA) revealed that the isolate was susceptible to ampicillin-sulbactam, amikacin, aztreonam, cefazolin, ceftazidime, cefotaxime, cefepime, cefoxitin, ertapenem, gentamicin, levofloxacin, meropenem, piperacillin-tazobactam, tigecycline, and trimethoprim-sulfamethoxazole. We performed a multiplex polymerase chain reaction for K1/K2 capsular serotyping and examined the expression of seven virulence genes using primer sets against *magA* (*wzy*-like polymerase specific to K1 strain), K2 capsular serotype-specific *wzi* gene, *rmpA*, *entB*, *ybtS*, *kfu*, *iutA*, *mrkD*, and *allS* to determine the virulence of the isolate, as described previously [8]. The string test for hypermucoviscosity revealed positive results, and the isolate was the K1 capsular serotype, which was positive for all seven virulence genes tested.

## 3. Discussion

We report a case of severe pyomyositis and osteomyelitis in the lower left leg with a multifocal septic embolism throughout the body caused by hvKP originating from an anorectal abscess. The awareness of hvKP as a pathogen responsible for causing highly invasive infections is the result of a relatively recent discovery [7]. Although pyomyositis caused by *K. pneumoniae* is rare, it can lead to severe metastatic infections. This case highlights the importance of phenotyping and genotyping in *K. pneumoniae* infections. We hope that increasing awareness of this category of infection will result in more prompt detections of metastatic infections and better patient management and outcomes.

HvKP is most commonly a community-acquired infection, and its antimicrobial susceptibility patterns remain largely pan-sensitive, with resistance usually observed only against ampicillin [9,10]. The *K. pneumoniae* isolated from this patient was susceptible to all the antimicrobials tested, except ampicillin. Although diabetes has been identified as a risk factor for hvKP infection, many patients, including the current patient, have no underlying immunodeficiencies or comorbidities [3,11,12]. The incidence of multidrug-resistant hvKP has been increasing. Li, et al. [13] found up to 60% resistance against third-generation cephalosporins and fluoroquinolones, and Zao et al. [14] reported an outbreak of carbapenem-resistant hvKP. As antimicrobial resistance will eventually elevate the threat level of hvKP, the risk of severe metastatic infections and the possibility of encountering a case of multidrug resistance should be considered from an early stage in patients with *K. pneumoniae*.

A few studies have focused on pyomyositis caused by *K. pneumoniae*, with one case of a K1 serotype harboring *rmpA* and *rmpA2*; [15] one case of capsular serotype K2 harboring *rmpA2*, *iutA*, *terW*, and *silS*; [11] and one case of serotype K2 harboring *rmpA*, *rmpA2*, aerobactin, and salmochelin) [16]. Each isolate demonstrated hypermucoviscosity, as evidenced by a positive string test. In our case, capsular serotype K1 was present, resulting in a positive string test. The hypervirulent phenotype was related to multiple virulent genes, including *rmpA, iutA, ybtS*, *entB*, *mrkD*, *kfu*, and *allS*. Further studies are required to determine the role of these virulence factors in the pathogenesis of pyomyositis and osteomyelitis.

The string test is sometimes used to distinguish hypervirulent from classic *K. pneumoniae*. However, many cases of a positive string test on *K. pneumoniae* isolates do not exhibit hypervirulence, and it remains unclear whether all hypervirulent *K. pneumoniae* are hypermucoviscous [5]. The hypervirulence of *K. pneumoniae* can be defined as the ability of the organism to cause invasive infections through metastatic dissemination from a primary infection site in healthy adults [17]. Various virulence factors have been reported to be associated with a hypervirulent phenotype [7]. Hypervirulence is not regarded as arising from a single gene but rather as a result of complex interactions of multiple genetic determinants [17]. In this isolate, various virulent factors, such as *iutA* and *rmpA*, were detected, and metastatic infection from the anorectal abscess as the primary source resulted in extensive pyomyositis, metastatic brain and renal abscesses, and osteomyelitis in the lower left leg. These findings suggest that the isolate in this study was a hypervirulent strain.

The Invasive syndrome caused by *K. pneumoniae* has been clinically defined as a pyogenic liver abscess with an extrahepatic metastatic infection [18]. However, in our patient, the primary infection site was not a pyogenic liver abscess but rather an anorectal abscess. Liver abscesses are likely initiated by a break in host defenses in the gastrointestinal tract, which is the reservoir or dominant site of colonization by *K. pneumoniae* that permits it to seed another site [19,20,21,22]. Anorectal abscesses in the gastrointestinal tract, similar to abscesses appearing in the liver, may be the primary sites of infection by *K. pneumoniae*. We believe that the observed metastatic complications occurred in this patient based on the connections between the anorectal abscess and the patient’s brain, lungs, kidneys, muscle, and bones. Ultimately, the prostate, kidneys, colon, and liver must be considered potential locations for a primary abscess caused by hvKP. Regardless of the location, they can cause metastatic infections. Therefore, when *K. pneumoniae* is identified in the presence of metastatic infections, such as endophthalmitis, pyomyositis, osteomyelitis, or a septic lung, the primary abscess location should be investigated in the absence of a liver abscess, and active drainage should be performed. In addition, we believe that the virulence profile of *K. pneumoniae* and the prompt surveillance of metastatic infections could result in better management and outcomes in patients.

## 4. Conclusions

Consistent with previous reports, the hvKP infection in this study was associated with high morbidity and severe physical disability. To provide optimal care and minimize undesirable consequences, clinicians must be prepared to quickly detect this pathogen using a virulence test, even if there is no evidence of a liver abscess. They should also consider the possibility of severe sepsis and multiple metastatic hvKP infections and be prepared with prompt surveillance and management, such as active drainage, to improve the patient’s outcome.

## Figures and Tables

**Figure 1 medicina-58-01450-f001:**
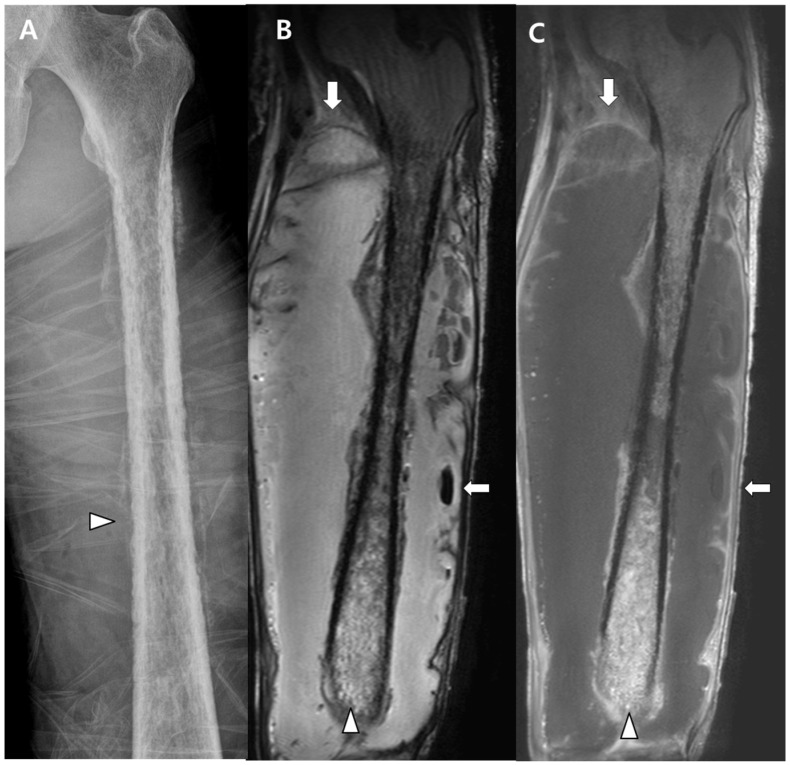
(**A**) Radiography of the left lower extremity shows periosteal thickening with osteolysis (arrowheads), which is compatible with osteomyelitis. (**B**,**C**) Coronal T2-weighted and T1 contrast enhancement MRI scans show extensive soft tissue abscess (arrows) with associated abnormal marrow signal intensities (arrowheads) representing pyomyositis with osteomyelitis.

**Figure 2 medicina-58-01450-f002:**
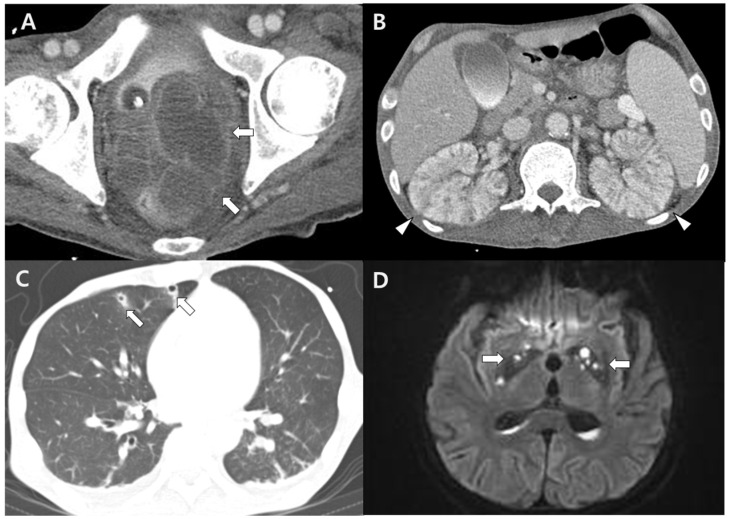
CT and MRI images show multiple metastatic infections. (**A**,**B**) Contrast-enhanced abdomen CT scan revealed multiple abscess pockets in the anorectal area (arrows) and microabscesses in both kidneys (arrowheads). (**C**) Chest CT scan showing multiple cavities representing septic embolism (arrows), and (**D**) brain diffusion MRI showing multifocal septic infarction in the basal ganglia (arrows).

## Data Availability

Not applicable.

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
