# Peer review of "Invasive Hypervirulent Klebsiella pneumoniae Syndrome Originating from an Anorectal Abscess as Opposed to a Pyogenic Liver Abscess"

_medicina, 2022, doi:10.3390/medicina58101450_

Round 1

Reviewer 1 Report

The increase number of infection caused by invasive hypervirulent Klebsiella pneumoniae is very important problem worldwide.

Therefore, the case report about hvKP is meaningful for clinicians. Here, I suggest a few revisions so that the manuscript adheres to the journal’s guidelines better.

1.    How about the result of blood culture?

2.    The authors should describe the reason why antibiotics were began to be administered at day 2 hospitalization.

3.    The authors had better to describe the reasons why the patient was immunocompetent not only with past history but also with screening of immunodeficiency.

Author Response

The increase number of infection caused by invasive hypervirulent Klebsiella pneumoniae is very important problem worldwide.

Therefore, the case report about hvKP is meaningful for clinicians. Here, I suggest a few revisions so that the manuscript adheres to the journal’s guidelines better.

  1. How about the result of blood culture?

â–¶ Response: Blood cultures at the ER visit and during hospitalization were negative. We added this sentence. (page 3)

  1. The authors should describe the reason why antibiotics were began to be administered at day 2 hospitalization.

â–¶ Response: Empirical antibiotics were started on the first day of admission. The position of the sentence pointed out by the reviewer has been changed. (page 2)

  1. The authors had better to describe the reasons why the patient was immunocompetent not only with past history but also with screening of immunodeficiency.

â–¶ Response: We presented the results of the immunoglobulin test, complement test, and HIV antibody test performed during the hospitalization period. (page 2)

Reviewer 2 Report

1-it is preferred to increase the section of introduction.

2- in section of introduction 7th line modify Klebsiella pneumoniae

 to K. pneumoniae.

3-section of conclusion is missing.

Author Response

1-it is preferred to increase the section of introduction.

â–¶ Response: We added the contents in the section of Introduction. (page 1-2)

2- in section of introduction 7th line modify Klebsiella pneumoniae

 to K. pneumoniae.

â–¶ Response: We modified “Klebsiella pneumoniae” to “K. pneumoniae”. (page 1)

3-section of conclusion is missing.

â–¶ Response: We made the section of conclusion. (page 5)

Round 2

Reviewer 2 Report

I recommend publishing manuscript